# Revealing the Hidden Impacts: Insights into Biological Aging and Long-Term Effects in Pauci- and Asymptomatic COVID-19 Healthcare Workers

**DOI:** 10.3390/ijms25158056

**Published:** 2024-07-24

**Authors:** Manuela Campisi, Luana Cannella, Anna Bordin, Angelo Moretto, Maria Luisa Scapellato, Paola Mason, Filippo Liviero, Sofia Pavanello

**Affiliations:** 1Department of Cardiac-, -Thoracic-, Vascular Sciences and Public Health, University of Padua, 35128 Padua, Italy; manuela.campisi@unipd.it (M.C.); luana.cannella@phd.unipd.it (L.C.); anna.bordin@unipd.it (A.B.); angelo.moretto@unipd.it (A.M.); marialuisa.scapellato@unipd.it (M.L.S.); paola.mason.1@unipd.it (P.M.); filippo.liviero@unipd.it (F.L.); 2Occupational Medicine, University Hospital of Padua, 35128 Padua, Italy

**Keywords:** biological aging, DNA methylation age, telomere length, post-COVID-19, healthcare workers, paucisymptomatic, heart rate variability, respiratory function, nasal cells, induced sputum

## Abstract

This study explores the role of inflammation and oxidative stress, hallmarks of COVID-19, in accelerating cellular biological aging. We investigated early molecular markers—DNA methylation age (DNAmAge) and telomere length (TL)—in blood leukocytes, nasal cells (NCs), and induced sputum (IS) one year post-infection in pauci- and asymptomatic healthcare workers (HCWs) infected during the first pandemic wave (February–May 2020), compared to COPD patients, model for “aged lung”. Data from questionnaires, Work Ability Index (WAI), blood analyses, autonomic cardiac balance assessments, heart rate variability (HRV), and pulmonary function tests were collected. Elevated leukocyte DNAmAge significantly correlated with advancing age, male sex, daytime work, and an aged phenotype characterized by chronic diseases, elevated LDL and glycemia levels, medications affecting HRV, and declines in lung function, WAI, lymphocyte count, hemoglobin levels, and HRV (*p* < 0.05). Increasing age, LDL levels, job positions involving intensive patient contact, and higher leukocyte counts collectively contributed to shortened leukocyte TL (*p* < 0.05). Notably, HCWs exhibited accelerated biological aging in IS cells compared to both blood leukocytes (*p* ≤ 0.05) and NCs (*p* < 0.001) and were biologically older than COPD patients (*p* < 0.05). These findings suggest the need to monitor aging in pauci- and asymptomatic COVID-19 survivors, who represent the majority of the general population.

## 1. Introduction

Nearly four years after the World Health Organization (WHO) declared COVID-19 a pandemic, with over 775 million confirmed cases by May 2024 [1], the long-term consequences of the disease on the health of those infected remain largely unknown and are a significant area of research for global health [2]. The term “Long COVID” describes symptoms that persist or develop after the acute phase of COVID-19. This includes symptoms that last from 4 to 12 weeks after the acute phase, as well as post-COVID-19 syndrome, which refers to symptoms lasting more than 12 weeks and not explained by an alternative diagnosis [3]. 

While extensive research has focused on post-COVID-19 conditions in severely ill patients [4], there has been less attention on those with mild or asymptomatic infections, such as many healthcare workers (HCWs). Of 81 studies on post-COVID prevalence symptoms up to one year after infection [5,6,7], only 8 studies focused on pauci- and asymptomatic subjects after one year [8,9,10,11,12,13,14,15]. Furthermore, despite mild initial symptoms, 28% to 76% of these individuals developed post-COVID-19 syndrome. A one-year follow-up study on a mixed population (hospitalized and non-hospitalized patients) conducted by Lombardo et al. [13] highlighted that more severe impairment in the acute phase did not appear to predict more serious complications. This gap is particularly important as these individuals represent a large portion of the population and underscore the importance of monitoring and supporting all COVID-19 patients, regardless of the initial severity of their symptoms, to adequately manage long-term complications. 

Inflammation and oxidative stress, hallmarks of COVID-19 disease, play a key role in cell biological aging, supporting the hypothesis of its acceleration in COVID-19 [16]. At the cellular level, two interconnected “pillars of aging” are the earliest targets of cellular aging, i.e., telomere length (TL) and DNA methylation age (DNAmAge) [17,18]. One study reported evidence of biological age acceleration (i.e., epigenetic age acceleration and telomere shortening) in severe COVID-19 patients [16] as well as in COVID-19 survivors [19], whereas Franzen et al. [20] reported no epigenetic age acceleration in COVID-19 patients. Furthermore, in our recent work, we demonstrate that the lung becomes older than blood, as measured by both TL and DNAmAge in induced sputum (IS) cells from the deep airways, compared to circulating blood leukocytes in the same COPD patients, chronically exposed to inflammatory injury [21]. COVID-19 infection and the consequent pulmonary oxidative–inflammatory reaction lead to structural and functional pathological changes in the lung and also postulate accelerated lung aging. To date, no one has investigated biological age indicators in pauci- and asymptomatic COVID-19 patients in different tissues other than blood, such as nasal cells (NCs) and IS cells from the deep airways, which are tissues preferentially infected by SARS-CoV-2. 

The aim of this study is to assess the biological aging of blood leukocytes and target tissues of the infection (i.e., IS and NCs) in SARS-CoV-2-positive HCWs of the first wave (February–May 2020) recruited at the health surveillance visit of approximately 12 months after infection and to verify the long-term sequelae of the infection and the impact on work capacity. 

Given the global impact of the COVID-19 pandemic, studying the potential influence of SARS-CoV-2 infection on accelerated biological aging is of significant public health, economic, and social relevance. For the first time, we are examining this effect not only in the blood but also in the target tissues primarily exposed to the virus. This research will enable the development of personalized strategies to facilitate a full return to work.

## 2. Results

### 2.1. Post-COVID Syndrome (PCS) and Symptom Prevalence 

In Table 1, the prevalence of PCS symptoms in the HCWs cohort (*n* = 76) at 12 weeks was higher in women than men (*p* = 0.0043) but similar after 1 year (*p* = 0.5238). However, after 1 year, symptoms decreased in women (*p* = 0.0204) while persisting in men (*p* = 0.9999). Figure 1 shows the percentage distribution of persistent COVID-19 symptoms reported by the HCWs cohort (*n* = 76) up to 4 weeks, from 4 to 12 weeks, beyond 12 weeks after diagnosis (NICE guidelines [22]), and at the 1-year follow-up. Dyspnea, palpitations, peripheral neuropathy, loss of concentration, memory problems, and anxiety, as well as rare symptoms like dermatological signs, persisted beyond 12 weeks and at the 1-year follow-up (*p* > 0.05). However, brain fog, sleep disorders, depression, and less frequent symptoms, including ocular symptoms, persisted beyond 12 weeks but not after 1 year (*p* > 0.05). All data are presented in Appendix A. Appendix A shows that in 30% of HCWs, symptoms persisted 1 year after SARS-CoV-2 infection. 

### 2.2. Blood Leukocyte Biological Age 

The mean values and standard deviations of blood leukocyte DNAmAge, AgeAcc, and TL for all HCWs (*n* = 76) are reported in Appendix A. Simple linear regression analyses confirmed that blood leukocyte DNAmAge positively correlated with chronological age (Figure 2A, r = 0.9433, *p* < 0.0001), and blood leukocyte TL negatively correlated with chronological age (Figure 2B, r = −0.3217, *p* = 0.0046). Increased blood leukocyte DNAmAge, but not TL, was associated with the duration of COVID-19 infection (Figure 3A, r = 0.0618, *p* = 0.596; Figure 3B, r = 0.2378, *p* = 0.0386). Additionally, subjects with greater biological age, detected by DNAmAge and TL, showed lower WAI (Figure 4A, DNAmAge r = −0.5169, *p* < 0.0001; Figure 4B, TL r = 0.2828, *p* = 0.0194).

### 2.3. Determinants of Blood Leukocyte DNAmAge and TL

Multiple linear regression analysis (Table 2) indicated that an increase in blood leukocyte DNAmAge was determined by age (*p* < 0.0001), being male (*p* = 0.014), presence of chronic diseases (*p* = 0.029), decline in lung function (FEV1, *p* = 0.0014), and decrease in lymphocyte count (*p* = 0.002). Decreased blood leukocyte TL was determined by age (*p* = 0.003) and reduced lymphocyte count (*p* = 0.033).

Multiple regression analysis (Table 3) showed that an increase in blood leukocyte DNAmAge correlated with a decrease in WAI (*p* = 0.0015) and daytime work (*p* = 0.0325) but not job position (*p* = 0.4352). Blood leukocyte TL decreased with job positions involving direct patient contact (healthcare assistants, nurses, doctors, residents) (*p* = 0.0295) but not with WAI (*p* = 0.2268) or daytime work (*p* = 0.1864).

Multiple regression analysis of the influence of hemoglobin (g/dL), glycemia (mg/dL), cholesterol (mg/dL), triglycerides (mg/dL), HDL (mg/dL), LDL (mg/dL), creatinine (mg/dL), and bilirubin (mg/dL) on blood leukocyte DNAmAge and TL (Table 4) revealed that higher DNAmAge was associated with lower hemoglobin (*p* = 0.0163), higher glycemia (*p* = 0.0078), and higher LDL (*p* = 0.0015). Shorter TL was associated only with higher LDL levels (*p* = 0.0506). Cholesterol, triglycerides, HDL, creatinine, and bilirubin were not determinants of biological aging indicators.

Multiple regression analysis of the influence of mean HR and HRV parameters (i.e., SDNN, RMSSD) and drugs affecting HRV (i.e., antidepressants, beta-blockers, calcium channel blockers, inhaled or oral beta-mimetics, theophylline, and alpha-adrenergic agonists) on blood leukocyte DNAmAge and TL (Table 5) indicated that increased DNAmAge correlated with low mean HR and drug use affecting HRV but not with other HRV parameters. No significant correlations were found for TL.

Multiple linear regression analysis of the influence of leukocytes (10^9^/L) and different blood cell counts, including neutrophils (10^9^/L), lymphocytes (10^9^/L), and monocytes (10^9^/L), on blood leukocyte DNAmAge and TL (Appendix A) showed positive correlations between TL and neutrophils (*p* = 0.0006) and lymphocytes (*p* = 0.0046) and negative correlations with leukocytes (*p* = 0.0344) but not with monocytes. DNAmAge was not determined by leukocytes, neutrophils, lymphocytes, or monocytes (*p* > 0.05).

### 2.4. Biological Age of Blood Leukocytes, IS Cells, and NCs

Appendix A reports the number of subjects and mean values of biological aging indicators, i.e., DNAmAge, AgeAcc, and TL, in blood leukocytes, NCs, and IS cells. DNAmAge of blood leukocytes (Figure 2A, r = 0.9433, *p* < 0.0001), NCs, and IS cells (Appendix A, A, r = 0.8015, *p* < 0.0001; B, r = 0.9279, *p* < 0.0001) was positively correlated with chronological age, while TL of blood leukocytes (Figure 2B, r = −0.3217, *p* = 0.0046), but not of IS cells (Appendix AC, r = −0.2641, *p* = 0.2897), was negatively correlated with chronological age. Insufficient DNA prevented TL analysis in NC samples.

Figure 5A shows that in a subset of 17 HCWs with all tissue samples, the IS DNAmAge was higher than blood leukocytes (*p* = 0.0011) and NCs (*p* = 0.0003), and NC DNAmAge was lower than blood leukocytes (*p* = 0.0028). Similarly, Figure 5B reports that IS TL was shorter than blood leukocytes in the same patients (*p* = 0.05).

### 2.5. Correlations between Biological Aging Indicators

Simple linear regression analyses (Appendix A) showed positive correlations between DNAmAge of blood leukocytes and NCs (A, r = 0.8207; *p* < 0.0001) and IS cells (B, r = 0.9353; *p* < 0.0001) but no correlation between TL of blood leukocytes and IS cells (C, r = −0.0222; *p* = 0.9304).

### 2.6. Comparison of Biological Aging in HCWs and COPD Patients

Comparing HCWs (*n* = 17) with COPD patients (*n* = 7), HCWs showed greater biological aging in blood and IS cells, including higher AgeAcc (Table 6: blood leukocyte AgeAcc, *p* = 0.0002; IS AgeAcc, *p* = 0.012) and shorter TL (Table 6: predicted blood and IS TL, *p* < 0.0001) one year after SARS-CoV-2 infection resolution.

## 3. Discussion

This study offers groundbreaking insights into the biological aging, long-term sequelae, and their impact on work ability in pauci- and asymptomatic HCWs of the University Hospital of Padua who survived COVID-19 during the first wave (February–May 2020). We also assessed the biological aging of blood leukocytes, NCs, and IS cells in HCWs and compared them to a control group of COPD patients, who are considered a model group for “aged lung” resulting from continuous inflammatory action affecting these patients [23].

### 3.1. PCS and Symptom Prevalence

At 12 weeks, PCS symptoms were more prevalent in women than men, consistent with Ortona et al.’s findings that women are twice as likely as men to develop PCS until around the age of 60 [24]. Given our population’s median age of 47.5 years and a peak age of 66 years, this trend is confirmed. Women make up about 71% of our study population and dominate the healthcare workforce, with the WHO reporting that women constitute 67% of the global health and social care workforce [25]. This demographic representation explains the higher prevalence of PCS among women in our study. Furthermore, the autoimmune hypothesis posits that women’s stronger immune response, influenced by genetic and hormonal factors, leads to a higher incidence of PCS [24,26]. After 1 year, symptom prevalence was similar between genders, with symptoms decreased in women but persisted in men. This persistence is an ongoing area of investigation. Overall, 30% of HCWs reported persistent symptoms 1 year post-infection. This is in line with previous studies in paucisymptomatic individuals, reporting similar rates at one year [11,27,28] and even two years post-infection [29]. Persistent symptoms such as dyspnea, palpitations, peripheral neuropathy, anxiety, loss of concentration, and memory problems were observed approximately one year after infection, confirming earlier findings [14,27]. These underscore the need for ongoing research and tailored healthcare strategies to manage the long-term effects of PCS, particularly among HCWs, who represent our study cohort.

### 3.2. Determinants of Increased Blood Leukocyte DNAmAge

#### 3.2.1. Sex-Related DNAmAge Differences

Male sex is significantly associated with increased DNAmAge, corroborating findings from Oblak et al. [30]. This aligns with the male–female health survival paradox, where males typically have shorter lifespans than females, who tend to experience higher rates of disability and poor health [31], again supporting the need for sex-specific health strategies. 

#### 3.2.2. Impact of SARS-CoV-2 Infection

Increased leukocyte DNAmAge correlates with the duration of SARS-CoV-2 infection (average 17 days) because prolonged infections lead to sustained inflammatory responses and cellular stress, which induce significant epigenetic changes [32,33,34,35]. This mechanism is similar to that observed in other viral infections like HIV [36,37,38] and SARS-CoV-2 infection [16], where extended viral presence exacerbates epigenetic modifications, thus increasing DNAmAge. This insight is crucial for developing post-infection management strategies, particularly for those with extended infection durations, to mitigate accelerated aging effects.

#### 3.2.3. Chronic Diseases and DNAmAge

Our findings confirm that chronic diseases are linked to elevated DNAmAge, consistent with previous research on frailty [39], cancer [40], diabetes [41], cardiovascular diseases (CVDs) [42], dementia [43], and decreased lung function (FEV1) in COPD patients [21], a known consequence of aging [44,45,46]. This underscores the importance of managing chronic diseases to potentially slow down the biological aging process and improve overall health outcomes.

#### 3.2.4. Lung Function and DNAmAge

We also found a correlation between the acceleration in DNAmAge and the decline in lung function measured by forced expiratory volume (FEV1), which is a well-documented consequence of aging [44,45,46]. This finding is consistent with our previous work on COPD patients [21]. Therefore, DNAmAge appears to be a reliable signature of the epigenetic aging chronic disease-related.

#### 3.2.5. Lipid Levels and DNAmAge

We identified a positive association between DNAmAge and LDL levels. Even if our finding contrasts with an unexpected inverse association reported by Ammous et al. [47], it aligns with the hypothesis on the detrimental health effects of these lipids [48,49], which are connected to an increased risk of CVD [50,51,52], and suggests that lipid management could be integral to mitigating accelerated aging. This highlights the need for comprehensive lipid monitoring and management strategies in PCS patients.

#### 3.2.6. Blood Glucose and DNAmAge

Our study also found a positive correlation between DNAmAge and blood glucose, which showed consistent and stronger associations with CVD risk factors in patients with diabetes (Borg et al., 2011). This correlation underscores the critical role of glucose regulation in aging. High blood glucose contributes to oxidative stress and inflammation, leading to epigenetic changes that increase DNAmAge. Effective glycemic control is crucial for slowing biological aging, particularly in diabetic populations, as highlighted by recent research [53].

#### 3.2.7. Work Capacity and DNAmAge

The higher DNAmAge found in daytime workers may be attributed to older age, as they transition to daytime roles due to the challenges of night shifts. Furthermore, HCWs with greater DNAmAge showed lower WAI scores, marking this as the first study to link leukocyte DNAmAge with WAI, consistent with the decline in work capacity due to aging and chronic diseases [54,55]. Work capacity, measured by WAI, refers to an individual’s perception of the balance between work demands and their ability to cope with them [56], resulting from the interaction between psychosocial and physical work-related elements, mental and physical capabilities, and health issues [57,58]. Our finding calls for workplace interventions to support aging workers, such as job modifications and health promotion programs, to sustain their work ability and productivity.

#### 3.2.8. Lymphocyte Counts and DNAmAge

We observed a negative correlation between DNAmAge and lymphocyte counts. Aging is known to reduce the number of B cells and CD4+ and CD8+ T cells, increasing memory T cells while decreasing naïve T cells [59]. This finding is supported by Zhang et al. [60], who noted variations in epigenetic age depending on lymphocyte subpopulations. Strategies to also support immune health could play a role in mitigating DNAmAge increases, especially in the aging workforce

#### 3.2.9. Hemoglobin Levels and DNAmAge

An association between higher DNAmAge and lower hemoglobin levels was found, reflecting the intertwined nature of aging and hematological health. Lower hemoglobin levels are indicative of anemia, which is common in older adults and linked to increased biological aging. This condition is often linked to increased biological aging due to its association with chronic inflammation, oxidative stress, and decreased erythropoiesis [61]. Monitoring and addressing hemoglobin levels can be beneficial in managing age-related health risks and potentially mitigating accelerated biological aging.

#### 3.2.10. HR, HRV, and DNAmAge

We found a relationship between increased DNAmAge and low mean HR. Basal and non-basal HR decreases with age, and elderly people are more prone to bradycardia [62], and even HRV decreases with age [63]. DNAmAge is a marker of aging associated with pathological conditions [64]. Increased DNAmAge signifies accelerated biological aging, exacerbated by factors like chronic stress and infections. COVID-19 survivors show increased DNAmAge [19] and reduced mean HR one year after SARS-CoV-2 infection compared to the post-acute phase [65], highlighting the impact of infections on biological aging. Maintaining HR and HRV through interventions could mitigate accelerated aging.

Furthermore, we noted an association between drugs affecting HRV and higher leukocyte DNAmAge. Drugs such as beta-blockers, calcium channel blockers, inhaled or oral beta-mimetics, theophylline, and alpha-adrenergic agonists like phenylephrine infusion can interfere with HRV [65,66]. While limited research exists, some studies indicate antihypertensives like calcium channel blockers may reduce DNAmAge [67]. Therefore, our findings highlight the need for further investigation into the effects of other HRV-modifying drugs on DNAmAge and elucidate the underlying biological mechanisms. Further research is necessary to elucidate the biological mechanisms and optimize pharmacological strategies to minimize adverse effects on biological aging.

Lastly, we established the robust positive correlation between DNAmAge and chronological age, reinforcing the accuracy of our analysis and the predictive strength of our model. This consistency with established methodologies [68,69] underscores that DNAmAge is a reliable biomarker for biological aging.

### 3.3. Determinants of Shorter Blood Leukocyte TL

Our study established a negative correlation between leukocyte TL and chronological aging, consistent with the existing literature, such as the systematic review by Müezzinler et al. [70] across 124 cross-sectional studies, which reported a similar negative correlation (r = 0.3).

#### 3.3.1. WAI

HCWs with shorter leukocyte TL exhibit lower WAI, reflecting accelerated biological aging. Chronic job-related stress and inflammation accelerate telomere shortening, impairing cellular repair and function [71]. This relationship is biologically plausible as shorter TL indicates advanced cellular aging, which reduces physical and cognitive capacity, impacting work ability. The parallel with epigenetic age (DNAmAge) further supports this connection, highlighting the detrimental effects of occupational stress on aging markers and work capacity. Interventions to reduce stress could improve HCWs’ health and work performance.

#### 3.3.2. LDL Levels and Cardiovascular Disease 

Our research supports the well-documented association between shorter TL and CVD, alongside elevated LDL levels as a major risk factor [72]. The observed correlation between shorter leukocyte TL and higher LDL levels contributes another piece to this controversial area, aligning with some previous studies [73,74,75,76,77] while contradicting others [76,78,79,80]. This underscores the need for further investigation into the interplay between lipid levels and TL.

#### 3.3.3. Blood Leukocyte TL and Job Position

The observation that blood leukocyte TL is decreased in HCWs involved in direct patient contact (assistants, nurses, doctors, and residents) can be explained by a few key mechanisms. HCWs face high levels of chronic stress due to long hours, high workload, and emotional strain [81]. Chronic stress increases cortisol levels, which can lead to oxidative stress and inflammation. Elevated cortisol from chronic stress increases oxidative stress, producing reactive oxygen species (ROS) that damage cells and DNA, including telomeres [71]. Frequent exposure to pathogens (viruses and bacteria) in healthcare settings triggers immune responses, increasing leukocyte replication and further contributing to telomere shortening [82,83,84]. Overall, the combination of chronic stress, oxidative stress, inflammation, and frequent exposure to pathogens leads to accelerated telomere shortening in HCWs involved in direct patient contact.

#### 3.3.4. Lymphocyte Numbers 

We observed a relationship between TL shortening and a decrease in lymphocyte numbers. Although lymphocytes experience a faster rate of age-dependent TL shortening than granulocytes [85], recent studies have shown significant reductions in lymphocyte numbers among healthy COVID-19 survivors [86]. Furthermore, our analysis revealed that TL shortening was associated with an increased total leukocyte count, primarily driven by monocytes. This is in line with findings of generalized low-grade inflammation, T lymphocyte senescence, and increased monocyte activation in individuals with long COVID [87]. Mean leukocyte TL is considered an indicator of biological aging [88]. Our findings, linking TL shortening to lymphocyte reduction, expand on this understanding, suggesting novel mechanisms underlying TL dynamics in relation to immune cell populations and chronic stress responses. 

These innovative results provide new insights into the determinants of leukocyte TL, advancing our comprehension of biological aging and its interaction with chronic disease, stress, and occupational factors.

### 3.4. Biological Age of the Blood Leukocytes, IS Cells, and NCs Determined by DNAmAge and TL

Our study uniquely analyzed DNAmAge and TL in blood leukocytes, IS cells, and NCs from the same cohort of 17 HCWs who survived COVID-19. This comprehensive comparison revealed that IS cells exhibit a higher biological age than both blood leukocytes and NCs. Specifically, IS cells demonstrated higher DNAmAge and shorter TL, while NCs had a lower DNAmAge compared to blood leukocytes, indicating different aging rates within the same individuals. 

#### 3.4.1. Tissue-Specific Aging Rates

The accelerated aging observed in IS cells compared to blood leukocytes and NCs confirms that different tissues and organs age at varying rates within the same individuals. This aligns with previous findings on heart, kidney, and COPD patients [21,89,90], showing tissue-specific aging rates [68]. The study also suggests that cells in the deeper airways of COVID-19 survivors are more susceptible to epigenetic changes than those in more superficial airways and blood leukocytes [21].

#### 3.4.2. COVID-19 Impact on DNAmAge and TL

Our study supports existing research showing that COVID-19 significantly alters DNA methylation profiles, particularly in critically ill patients [91,92]. These epigenetic changes persist even after recovery, suggesting long-term impacts on gene expression and cellular function [34,93,94,95]. Additionally, shorter telomeres are associated with severe COVID-19 and lingering post-COVID-19 conditions, supporting the hypothesis that telomere attrition plays a role in the pathology of COVID-19 [96,97,98]. Telomere shortening can lead to cellular aging and reduced regenerative capacity, contributing to the severe and long-lasting effects observed in COVID-19 patients [96,99,100]. Our findings agree with other studies that have documented accelerated biological aging in various tissues due to COVID-19 [16,19,101], emphasizing the need for further research into the long-term effects of the virus on cellular health and aging.

#### 3.4.3. Biological Implications of Telomere Shortening in IS Cells

Telomere shortening, a marker of biological aging, occurs with increased cell division and DNA replication. Our study found that IS cells, the primary target of SARS-CoV-2, showed significant telomere shortening, supporting previous findings in alveolar epithelial cells of COVID-19 patients [101]. This suggests that SARS-CoV-2 infection accelerates telomere shortening due to an enhanced proliferative response to regenerate alveolar injury, potentially leading to long-term lung fibrosis [101,102].

#### 3.4.4. Epigenetic Aging in IS Cells, NCs, and Implications for Surrogate Tissue Use

While the change in methylation profile is evident in the blood of COVID-19 patients and survivors [16,19,103], there are no data yet on the other target tissues. However, epigenetics, linking environmental and genetic factors [104], is recognized as the basis of inflammation [105], which underpins several lung diseases like COPD, cancer, and COVID-19 [106]. The lung is the primary target of SARS-CoV-2 infection, causing diffuse alveolar damage, apoptotic epithelial cells, interstitial inflammation, and activated T-cell responses resulting in a cytokine storm [107], mainly through host immune dysregulation, increased inflammation and/or hyperinflammation. This similarity and overlap in the pathogenetic mechanism between lung disease and COVID-19 [106], coupled with the results of our previous study in COPD patients showing accelerated lung aging [21], supports our finding that IS cells are older than blood leucocytes.

Interestingly, NCs were found to be biologically younger than IS cells and blood leukocytes. This could be due to their role as the initial entry point for SARS-CoV-2 [108], acting as a gateway to the lower respiratory tract and triggering systemic inflammation upon viral replication [109], or as a gateway to the central nervous system [110,111]. 

A strong correlation was found between the DNAmAge of IS cells, NCs, and blood leukocytes, suggesting that blood leukocytes could serve as a surrogate for studying lung and airway aging. However, caution is advised as there was a noted six-year difference in DNAmAge between lung tissue and blood leukocytes in post-COVID-19 patients, mirroring findings in COPD patients [21].

### 3.5. Comparison of Biological Aging (AgeAcc and TL) in HCWs and COPD Patients

Given the numerous overlaps between COPD and COVID-19 [106], we compared a group of HCWs recruited approximately one year after SARS-CoV-2 infection with a group of COPD patients as a positive control group. COPD patients are considered a suitable positive control group because they exemplify accelerated biological aging due to chronic inflammation and oxidative stress [21,112,113,114,115]. 

Our results revealed that the blood leukocytes and IS cells of HCWs are biologically older than those of COPD patients, as determined by AgeAcc and predicted TL. This indicates that COVID-19 may induce more pronounced epigenetic changes and telomere attrition than COPD. 

### 3.6. Limitations and Strengths

The current study has several limitations. Firstly, the lack of a control group of COVID-19-free, age-matched subjects is a significant limitation. This was due to the difficulty in recruiting HCWs approximately one year after infection during the first wave (February–May 2020), when workloads were high and reinfections were common, leading to the exclusion of these subjects from the study. To address this, we compared our HCWs population (*n* = 17), for whom we had all three tissues available, with a small positive control group of COPD patients (*n* = 7) known for accelerated lung aging compared to blood leukocytes [21,112,115]. We acknowledge that the sample size of our study is limited. However, this limitation arises from the low number of HCWs (*n* = 144 out of *n* = 8240 HCWs) infected during the first wave of COVID-19, which is a point of pride for our hospital. Additionally, the number was further reduced by the availability of healthcare workers who met the inclusion criteria and were willing to participate in the study (*n* = 76).

The ongoing pandemic and restrictions imposed by the University Hospital of Padua on techniques involving droplets and airborne exposure, such as spirometry and the IS technique, limited the number of IS samples collected. Despite the limited number of subjects (*n* = 17) for whom we have all three tissues (IS cells, NCs, and blood leukocytes), our sample size estimate indicates it is sufficient to obtain statistically significant results. Another limitation is the inability to analyze TL in NC samples due to insufficient DNA after performing DNAmAge analysis.

Our study has several strengths. Firstly, it provides a comprehensive assessment of biomarkers of biological aging, both genetic (TL) and epigenetic (DNAmAge), related to various parameters, including inflammation, basic hematochemical biomarkers, lung function indicators, and data on demographics, lifestyle, work, and physiological history. This study involved the collaboration of many healthcare professionals, including clinicians and researchers.

To our knowledge, this is the first study to determine the biological aging of post-COVID-19 subjects across three different tissues collected from the same individual, comparing blood leukocytes with SARS-CoV-2 target tissues (IS cells and NCs). We found a strong correlation between the DNAmAge of IS cells and NCs with that of blood leukocytes, suggesting the potential use of blood as a surrogate indicator of the biological age of IS cells and NCs, although further investigations are needed.

Another strength of our study is the use of a validated non-invasive airway sampling technique, the IS technique, to study biological age indicators in IS cells. This technique could be valuable for future research on lung biological aging, not only in COPD patients but also in other conditions.

## 4. Materials and Methods

### 4.1. Study Design

This study is a cross-sectional study in which the established clinical protocol has been applied to examine SARS-CoV-2-positive healthcare workers (HCWs) at the University Hospital of Padua approximately 1 year after the diagnosis of SARS-CoV-2 infection. The inclusion criteria were to be a SARS-CoV-2-positive HCW of the University Hospital of Padua, not have COVID-19 vaccination, and not have had COVID-19 reinfection in the 12 months preceding the clinical visit. The study population consisted of the *n* = 76 among the *n* = 144 HCWs of the first wave (February–May 2020) who met the inclusion criteria. They were enrolled during the health surveillance activity foreseen according to Legislative Decree 81/2008 at the Occupational Medicine Unit approximately 12 months after the diagnosis of SARS-CoV-2 infection with a molecular swab. Paucisymptomatic and asymptomatic HCWs, who did not have symptoms at the time of the swab and who continued to remain asymptomatic during the entire isolation period (at least 14 days), were included in the study population. The local Ethics Committee approved the study (288n/AO/22) in accordance with the principles of the Declaration of Helsinki.

All participants were informed of the purpose of the study and provided written informed consent. Subjects unwilling to participate in the study were excluded. At enrolment, each study participant was characterized by an ad hoc structured questionnaire to collect information on, among others, demographics, lifestyle, medical history, and environmental and occupational exposure. The assessment of medium–long-term effects in HCWs who had confirmed COVID-19, irrespective of whether they were hospitalized or had a positive SARS-CoV-2 molecular swab, was performed according to guidelines for managing the long-term effects of COVID-19 [116]. All patients underwent a clinical examination, including respiratory function tests, cardiac assessment with evaluation of heart rate variability (HRV), and administration of the Work Ability Index (WAI) questionnaire to assess their work ability. For each patient, the biological samples were also collected for basic biochemistry tests, immunological profiles, and biological aging analyses. Appendix A summarizes the characteristics of the study population, including demographic variables, lifestyle, occupational history, basic biochemistry parameters, liver function, inflammation, lung function, and HRV. Appendix A reports all data on the course of SARS-CoV-2 infection. The biological aging of HCWs was compared with that of a positive control group for biological aging parameters, including *n* = 7 patients with chronic obstructive pulmonary disease (COPD) who gave their final consent to participate in the study [21], which had already been approved by the Ethics Committee (3849/AO/16) in accordance with the principles of the Declaration of Helsinki.

### 4.2. Information Acquired through Questionnaires

An ad hoc structured questionnaire [117] was administered during interviews to elicit information on demographics (age, sex, marital status) and other personal information (mother/father age at birth, years of education), occupation [job title; hospital department; total years worked; years spent in the current job; shift work (work was considered scheduled in day shift from 6 a.m. to 2 p.m., afternoon shift from 2 p.m. to 8 p.m., and night shift from 8 p.m. to 6 a.m.); frequency of night shifts/month; job energy requirement (expressed as metabolic equivalent, MET) at work; work injury], and medically relevant complaints including cardiovascular disease, musculoskeletal disorder, spinal disc hernia, gastrointestinal disease, endocrine disease, diabetes, respiratory disease, and tumors. The Charlson comorbidity index, a method of predicting mortality by classifying or weighting comorbid conditions (comorbidities), was calculated, excluding diabetes, tumors and/or respiratory diseases, and other inflammatory conditions [118]. Smoking history (current active smokers, former smokers, and never smokers) and pack-years [(number of cigarettes smoked per day/20) × number of years smoked] were also recorded, as well as the habitual alcohol consumption (yes/no), alcohol intake (units of drink/day, each unit being approximately 10–12 g alcohol intake), and binge drinking (>4 drink-units/day, i.e., more than 40 g alcohol/day). Physical activity in leisure time was estimated according to the International Physical Activity Questionnaire (IPAQ score).

### 4.3. Work Ability Assessment

The Work Ability Index (WAI), a self-assessment questionnaire consisting of 7 domains, was used as previously described [118]. WAI ranged from 7 to 49 points; four categories were identified to describe WAI levels: “poor” (score 7–27), “moderate” (score 28–36), “good” (score 37–43), and “excellent” (score 44–49) work ability as a function of the total WAI score. This is a valuable tool to identify any imbalances between what is required (performance requirements) and what you are able to give (individual potential) [119].

### 4.4. Respiratory Function Tests

All lung function measurements, including forced expiratory volume in 1 s (FEV1), forced vital capacity (FVC), forced expiratory flows at different lung volumes, total lung capacity (TLC), and residual volume (RV), were measured using a spirometer (Jaeger MasterScreen PFT, PRO, Viasys Sanità, Firenze, Italy) according to the guidelines/recommendations of the American Thoracic Society/European Respiratory Society (ATS/ERS) [120]. Predicted normal values from the Communaute Europeenne du Carbon et de l’Acier (CECA) were used [121].

### 4.5. Assessment of Autonomic Cardiac Balance and HRV Parameters

ECG was recorded during periodical health checks. Blood pressure was also measured with an Omron 705IT electronic device (Omron Healthcare Europe, The Netherlands) while the patient was lying quietly for at least 5 min, according to the recommendations of the 2023 European Society of Hypertension [122]. Subjects were instructed to avoid smoking and to stop coffee and alcohol intake for 2 h and 48 h, respectively. They should have had sufficient (at least 8 h) rest and must not have worked the night shift before the test was performed. Blood pressure was measured once with a sphygmomanometer while the patient was lying calmly. HRV was assessed by an ECG performed in a supine position under physiologically stable conditions and using a device connected to the patient via two electrodes. HRV data were acquired by a Bluetooth acquisition system (BT16 Plus, FM, Monza, Italy). ECG was recorded for at least 5 min between 9 a.m. and 2 p.m., at rest and under ideal temperature conditions. HRV was analyzed using Kubios HRV software (ver. 3.3) [123]. Normal and aberrant complexes were identified, and all adjacent intervals between normal beats over 5 min intervals were considered. We analyzed the spectral components (HRV frequency domain variables) as the absolute values of power (ms2). Power spectral density was analyzed with an autoregressive modeling-based method (AR spectrum) using the default value for the model order, i.e., 16. The main spectral components were very low frequency (VLF), low frequency (LF), high frequency (HF), and the LF/HF ratio. The area under the curve of the spectral peaks within the frequencies 0.01–0.4, 0.01–0.04, 0.04–0.15, and 0.15–0.40 Hz were defined as the total power (TP), very low-frequency power (VLF), low-frequency power (LF), and high-frequency power (HF), respectively. In order to normalize LF and HF, we used the total power within the frequency range of 0.01–0.4 Hz. The normalized low-frequency power (nLF = LF/TP) corresponds to an index of combined sympathetic and vagal modulation [124] as well as a baroreflex index [125,126], while the normalized HF power (nHF = HF/TP) represents an index of vagal activity. The low-/high-frequency power ratio (LF/HF) is thus an index of sympathovagal balance. Time domain measures included the standard deviation of normal-to-normal RR intervals (SDNN) and the root mean square of successive RR interval differences (RMSSD).

### 4.6. Samples Collection and IS Procedure 

For each patient, blood samples were collected for basic biochemistry, immunological profile, and biological aging analyses (i.e., TL and DNAmAge). During medical examination, the procedures of sputum induction and nasal swabbing were carried out for each patient to collect both a sample of airway cells and nasal epithelium cells, respectively, for biological aging analyses. 

IS procedure was performed according to a standard protocol, and the IS sample was processed as previously described [21].

### 4.7. Basic Biochemistry Analyses

Data on basic biochemistry included number of blood red cells, platelets, white cells, lymphocytes, monocytes, neutrophils, basophils, eosinophils, hematocrit, hemoglobin, blood glucose, triglycerides, cholesterol, low-density lipoprotein (LDL), high-density lipoprotein (HDL), c-reactive protein (CRP), interleukin 6 (IL-6), alanine aminotransferase (ALT), aspartate aminotransferase (AST), gamma glutamyl transferase (gamma-GT), ferritin, total bilirubin, protein profile, and creatinine. All the analyses were performed at the Laboratory Medicine Unit (AOUP).

### 4.8. DNA Extraction (from Biological Samples)

DNA was extracted from whole blood samples using the QIAamp DNA Mini Kit (Qiagen, Milano, Italy) on a QIAcube System (Qiagen, Milano, Italy) for automated high-throughput DNA purification, according to a customized protocol as previously described [21]. 

DNA extraction was also carried out from the IS and nasal cells collected by the automated QIAcube System (Qiagen, Milano, Italy) utilizing QIAamp DNA Mini Kit (Qiagen, Milano, Italy) according to a customized protocol developed for highly viscous samples as previously described [21]. DNA was quantified and checked for quality using the QIAxpert Quantification System (Qiagen, Milano, Italy).

### 4.9. DNAmAge Analysis and AgeAcc Estimation

DNAmAge was determined by analyzing the methylation levels of five selected markers (ELOVL2, C1orf132, KLF14, TRIM59, and FHL2) in genomic DNA using bisulfite conversion and Pyrosequencing^®^ methodology on PyroMark Q48 Autoprep (QIAGEN, Milano, Italy), as previously described [127]. The methylation levels were expressed as a percentage of methylated cytosines at the 5 CpG sites considered and were used for the estimation of DNAmAge as previously reported [127]. All samples were analyzed 3 times for each marker to verify the reproducibility of our results, and their average was used in statistical analyses. All samples were analyzed on two different days, and the coefficient of variation (CV) in replicate Pyrosequencing runs was 0.5%. AgeAcc was computed as the discordance between the DNAmAge of blood leukocytes and IS cells and NCs and the subjects’ chronological age.

### 4.10. TL Analysis

TL was determined using quantitative real-time PCR after DNA extraction from both whole blood and IS samples [128]. This assay determines the ratio of telomere repeat copy number (T) to a single nuclear copy gene (S) in experimental samples relative to the T/S ratio of a reference pooled sample to determine measure TL in genomic DNA. Human (beta) globin (*hbg*) was the single-copy gene used. The PCR runs were performed in triplicate using a StepOnePlus Real-Time PCR System (Applied Biosystems, Milano, Italy), and the average of the three T/S ratio measurements was considered in the statistical analyses. To assess measurement reproducibility, 20% of samples were replicated on separate days, and the CV for the average T/S ratio was accepted if less than 10%.

### 4.11. Statistical Analyses

Univariate and multivariate methods were used to select the appropriate models. The analyses were performed using the statistical software StatsDirect and Rstudio. Regarding the analysis of biological age, our hypothesis of an accelerated lung and nasal epithelium caused by COVID-19 infection was converted into a model with two final outcomes: TL and DNAmAge, as previously described [21]. The biological aging of HCWs after 1 year from COVID-19 was compared to that of COPD patients as a positive control group for biological aging parameters. For this comparison, we used the AgeAcc, i.e., the difference between DNAmAge and chronological age, and TL predicted by regressing TL measurements on chronologic age for each subject.

## 5. Conclusions

This study provides comprehensive insights into the biological aging and long-term health impacts in pauci- and asymptomatic COVID-19 HCWs. By investigating persistent symptoms and early molecular markers such as DNAmAge and TL in blood leukocytes, NCs, and IS cells, several significant findings were revealed.

### 5.1. Persistent Symptoms

Despite mild or asymptomatic initial infections, a significant portion of HCWs experienced persistent symptoms such as dyspnea, palpitations, peripheral neuropathy, and cognitive issues up to one year after infection. This highlights the need for continuous monitoring and support for HCWs, as these long-term symptoms can affect overall health and work capacity.

### 5.2. Impact of SARS-CoV-2 Infection Duration on Biological Aging

Prolonged infection duration correlated with increased DNAmAge, suggesting that sustained inflammatory responses and cellular stress contribute to biological aging. This insight is crucial for developing targeted post-infection management strategies.

### 5.3. Key Determinants of Increased Biological Aging

Factors such as advancing age, male sex, presence of chronic diseases, daytime work, elevated LDL and glycemia levels, use of drugs affecting HRV, reduced lung function, a lower WAI and HR, and decreased lymphocyte count and hemoglobin levels were significantly associated with increased DNAmAge. Similarly, older age, increased LDL levels and leukocyte count, a job with direct patient contact, and a reduction in lymphocyte and neutrophil count were strongly associated with shorter TL. These findings underscore the complex interplay of biological and environmental factors in cellular aging and the importance of managing chronic conditions and maintaining healthy metabolic and cardiovascular profiles to mitigate accelerated aging.

### 5.4. Biological Aging in Target Tissues of SARS-CoV-2 Infection

The study identified accelerated biological aging in IS cells compared to blood leukocytes and NCs. This suggests that different tissues exhibit varying rates of aging after SARS-CoV-2 infection, with lung tissue (represented by IS cells) being more susceptible to accelerated aging.

### 5.5. Comparison with COPD Patients

The study found that HCWs exhibited greater biological aging in both blood leukocytes and IS cells compared to COPD patients. This suggests that COVID-19 may induce more significant epigenetic changes and telomere attrition than chronic inflammatory diseases like COPD.

In conclusion, this study’s innovative approach to assessing biological aging across multiple tissues reveals significant long-term impacts of SARS-CoV-2 infection. Our findings highlight that lung tissue is particularly affected, with HCWs showing accelerated aging in both blood leukocytes and IS cells compared to COPD patients. These results emphasize the need for continuous health monitoring, tailored management strategies, and supportive interventions for HCWs and pauci- and asymptomatic COVID-19 survivors, who represent a significant portion of the general population. Addressing the long-term consequences of the pandemic remains a critical public health priority. Further research is essential to fully understand COVID-19’s impact on biological aging and to develop effective strategies to mitigate these long-term health effects.

## Figures and Tables

**Figure 1 ijms-25-08056-f001:**
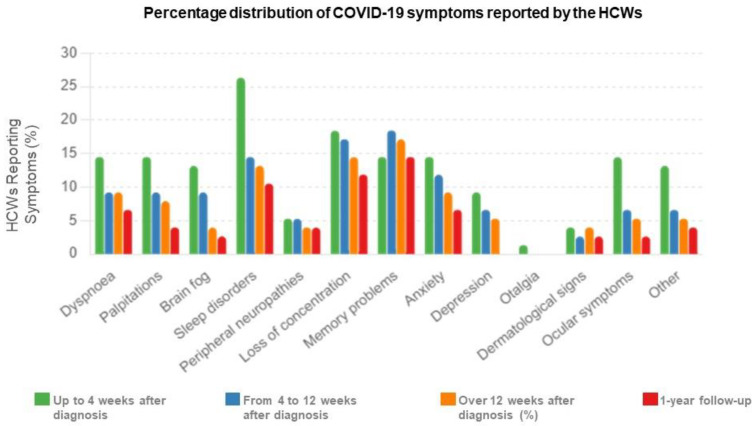
Percentage distribution of persistent COVID-19 symptoms reported by the HCWs up to 4 weeks, from 4 to 12 weeks, more than 12 weeks after diagnosis, and at the time of the visit (1-year follow-up) (*n* = 76 HCWs). The bar chart illustrates the percentage distribution of COVID-19 symptoms reported by HCWs up to 4 weeks (green), from 4 to 12 weeks (blue), more than 12 weeks after diagnosis (orange), and at the 1-year follow-up (red).

**Figure 2 ijms-25-08056-f002:**
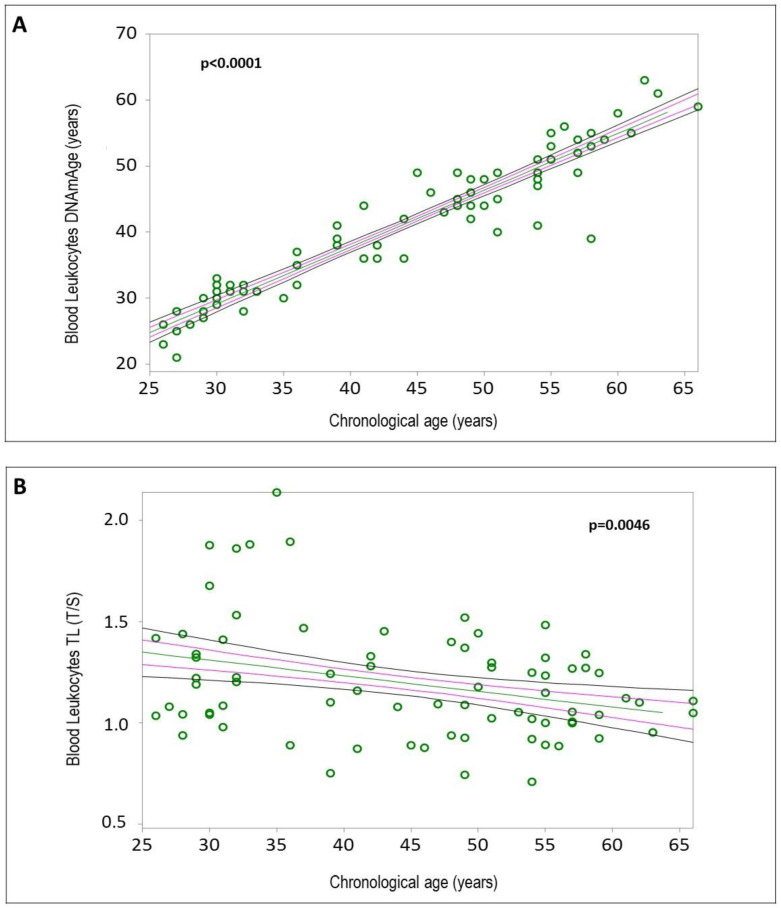
Correlation curves between blood leukocyte DNAmAge (**A**) or TL (**B**) and chronological age of *n*= 76 HCW COVID-19 survivors. In (**A**), a simple linear regression plot shows the correlation between blood leukocyte DNAmAge and chronological age [correlation coefficient (r) = 0.9433; two-sided *p* < 0.0001], while in (**B**), a simple linear regression plot shows the correlation between blood leukocyte TL and chronological age [correlation coefficient (r) = −0.3217; two-sided *p* = 0.0046]. Mean, standard error (SE), and 95% coefficient intervals (CIs) are represented as green, pink, and black lines, respectively. Values for each subject are shown as green circles.

**Figure 3 ijms-25-08056-f003:**
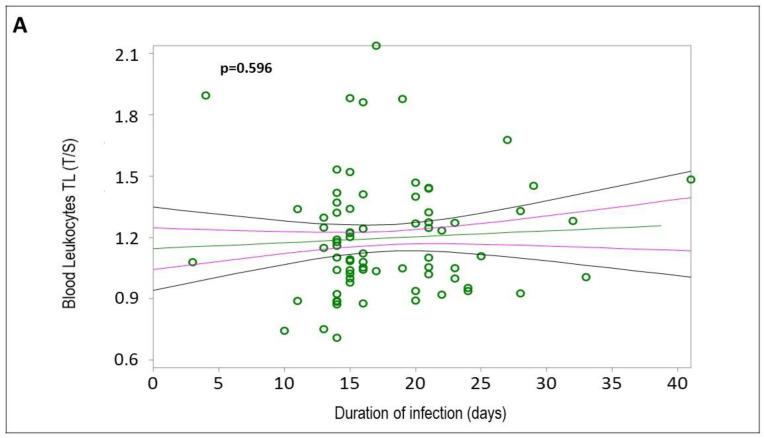
Correlation curves between blood leukocyte TL (**A**) or DNAmAge (**B**) and duration of infection (days) in *n* = 76 HCW COVID-19 survivors. In (**A**), a simple linear regression plot shows the correlation between blood leukocyte TL and days of infection [correlation coefficient (r) = 0.0618; two-sided *p* = 0.596], while in (**B**), a simple linear regression plot shows the correlation between blood leukocyte DNAmAge and days of infection [correlation coefficient (r) = 0.2378; two-sided *p* = 0.0386]. Mean, standard error (SE), and 95% coefficient intervals (CIs) are represented as green, pink, and black lines, respectively. Values for each subject are shown as green circles.

**Figure 4 ijms-25-08056-f004:**
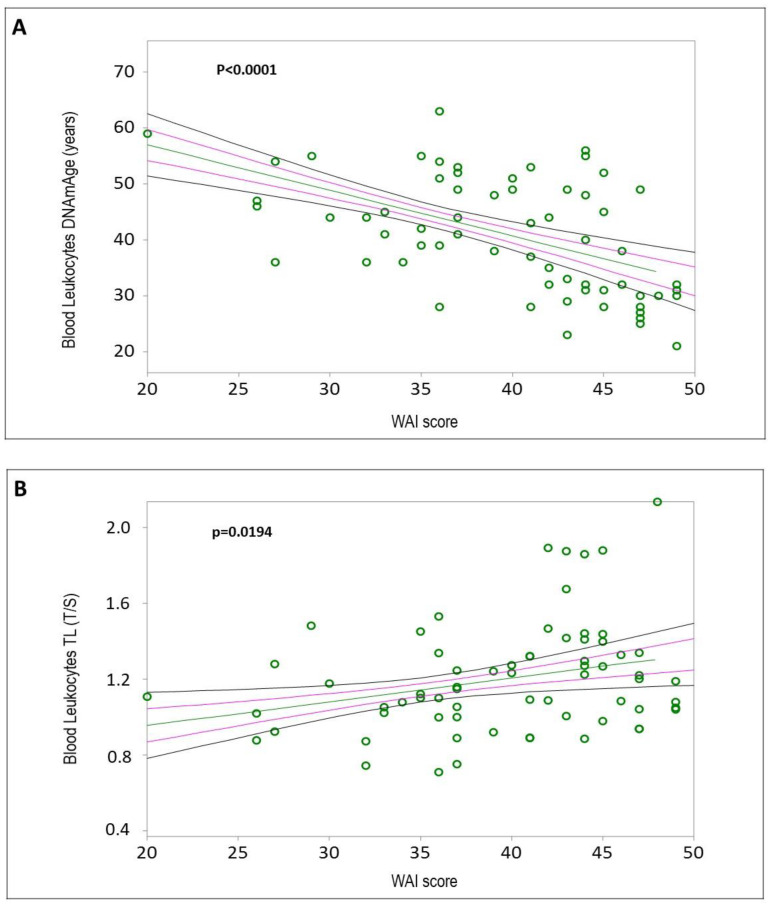
Correlation curves between blood leukocyte DNAmAge (**A**) or TL (**B**) and Work Ability Index (WAI) in *n* = 76 HCW COVID-19 survivors. In (**A**), a simple linear regression plot shows the correlation between blood leukocyte DNAmAge and WAI score [correlation coefficient (r) = 0.5169; two-sided *p* < 0.0001], while in (**B**), a simple linear regression plot shows the correlation between blood leukocyte TL and WAI score [correlation coefficient (r) = 0.2828; two-sided *p* = 0.0194]. Mean, standard error (SE), and 95% coefficient intervals (CIs) are represented as green, pink, and black lines, respectively. Values for each subject are shown as green circles.

**Figure 5 ijms-25-08056-f005:**
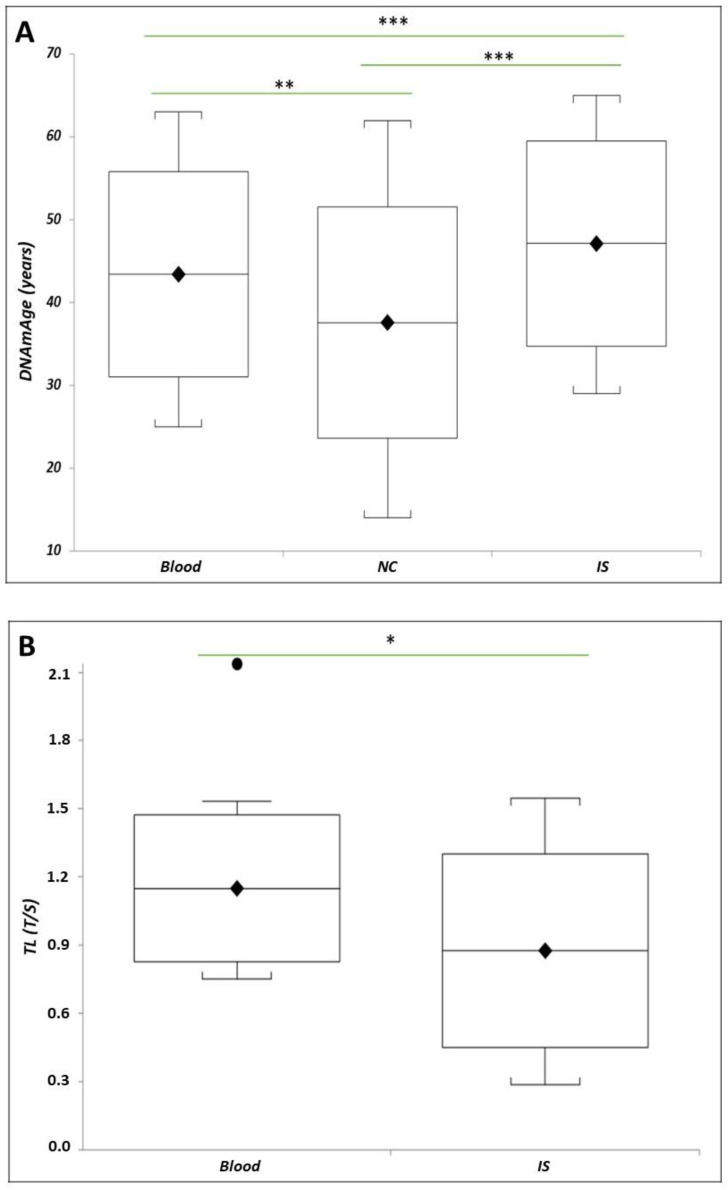
DNAmAge and TL of blood leukocytes, NCs, and IS cells in the subgroup of *n* = 17 HCW COVID-19 survivors. In (**A**), box plots show levels of DNAmAge (years) in blood leukocytes, NCs, and IS cells in the subgroup of *n* = 17 HCW COVID-19 survivors from whom all three tissue samples were collected. In box plots, the boundary of the box closest to the x-axis indicates the 25th percentile, the line within the box and the rhombus mark the mean, and the boundary of the box farthest from the x-axis indicates the 75th percentile. Whiskers (error bars) above and below the box indicate the 95 and 5th percentiles. The horizontal bar with asterisks indicates the significant comparison between blood leukocytes and paired NCs of the same subject (** paired *t*-test: mean 43.4 ± 12.9 years vs. mean 37.6 ± 14.0 years; *p* = 0.0028), NCs and paired IS cells of the same subject (*** paired *t*-test: mean 37.6 ± 14.0 years vs. mean 47.1 ± 12.4 years; *p* = 0.0003), and blood leukocytes and paired IS cells of the same subject (*** paired t-test: mean 43.4 ± 12.9 years vs. mean 47.1 ± 12.4 years; *p* = 0.0011). In (**B**), box plots show levels of TL in blood leukocytes and paired IS cells of *n* = 17 HCW COVID-19 survivors. In box plots, the boundary of the box closest to the x-axis indicates the 25th percentile, the line within the box and the rhombus mark the mean, and the boundary of the box farthest from the x-axis indicates the 75th percentile. Whiskers (error bars) above and below the box indicate the 95th and 5th percentiles. The black dot indicates an outlier. The horizontal bar with an asterisk indicates the significant comparison between blood leukocytes and paired IS cell TL of the same patient (* paired *t*-test (*n* = 7): mean 1.15 ± 0.32 T/S vs. mean 0.88 ± 0.43 T/S; *p* = 0.05).

**Table 1 ijms-25-08056-t001:** Prevalence of symptoms (post-COVID syndrome—PCS) in the HCWs cohort (*n* = 76) divided by sex and period since infection.

	PCS > 12 Weeks	PCS~1 Year	P_Trend
HCWs	0.4605	0.3026	0.0663
Women	0.5741	0.3333	**0.0204**
Men	0.1818	0.2273	0.9999
P_Trend	**0.0043**	0.5238	

**Table 2 ijms-25-08056-t002:** Multiple regression analysis of the influence of age, sex, presence of chronic diseases, decline in lung function (FEV1), and lymphopenia on blood leukocyte DNAmAge/LTL.

Blood Leukocyte DNAmAge (years)		**b**	**r**	**t Value**	** *p* **
	Age	b1 = 0.74399	r = 0.911647	t = 17.884209	**<0.0001**
	Sex (male)	b2 = 2.703646	r = 0.298937	t = 2.525593	**0.014**
	Chronic diseases (0 = no; 1 = yes)	b3 = 1.860584	r = 0.266824	t = 2.23213	**0.0291**
	FEV1 (L)	b4 = −2.219589	r = −0.382602	t = −3.338666	**0.0014**
	Lymphocytes (10^9^/L)	b5 = −2.017434	r = −0.36957	t = −3.206587	**0.0021**
Blood Leukocyte TL (T/S)	Age	b1 = −0.011446	r = −0.363521	t = −3.146031	**0.0025**
	Sex (male)	b2 = 0.004352	r = 0.005765	t = 0.046482	0.9631
	Chronic diseases (0 = no; 1 = yes)	b3 = 0.131573	r = 0.218459	t = 1.804868	0.0757
	FEV1 (L)	b4 = −0.023505	r = −0.05008	t = −0.404266	0.6873
	Lymphocytes (10^9^/L)	b5 = 0.120086	r = 0.261295	t = 2.182445	**0.0327**

Abbreviations: DNAmAge = DNA methylation age; FEV1 = forced expiratory volume in the 1st second; LTL = leukocyte telomere length.

**Table 3 ijms-25-08056-t003:** Multiple regression analysis of the influence of occupation or professional position, night shift work, and Work Ability Index (WAI) on blood leukocyte DNAmAge/TL.

Blood Leukocyte DNAmAge (years)		**b**	**r**	**t Value**	** *p* **
	Occupation (0 = HA; 1 = N; 2 = D; 3 = R; 4 = T and A)	b1 = −0.727891	r = −0.097697	t = −0.785337	0.4352
	Night shift work (0 = no; 1 = yes)	b2 = −5.367146	r = −0.263551	t = −2.185686	**0.0325**
	WAI	b3 = −0.617807	r = −0.382432	t = −3.311157	**0.0015**
Blood Leukocyte TL (T/S)	Occupation (0 = HA; 1 = N; 2 = D; 3 = R; 4 = T and A)	b1 = 0.064564	r = 0.268124	t = 2.226516	**0.0295**
	Night shift work (0 = no; 1 = yes)	b2 = 0.102616	r = 0.164683	t = 1.3357	0.1864
	WAI	b3 = 0.007125	r = 0.150818	t = 1.220504	0.2268

Abbreviations: HA = healthcare assistant, N = nurse, D = doctor, R = resident, T and A = technician and administrator.

**Table 4 ijms-25-08056-t004:** Multiple regression analysis of the influence of hemoglobin (g/dL), glycemia (mg/dL), cholesterol (mg/dL), triglycerides (mg/dL), HDL (mg/dL), LDL (mg/dL), creatinine (mg/dL), and bilirubin (mg/dL) on blood leukocyte DNAmAge/TL.

Blood Leukocyte DNAmAge (years)		**b**	**r**	**t Value**	** *p* **
	Hemoglobin (g/dL)	b1 = −0.205578	r = −0.300885	t = −2.484288	**0.0157**
	Glycaemia (mg/dL)	b2 = 0.2006	r = 0.329513	t = 2.748063	**0.0078**
	Cholesterol (mg/dL)	b3 = 0.006919	r = 0.022835	t = 0.179853	0.8579
	Triglycerides (mg/dL)	b4 = 0.003211	r = 0.01292	t = 0.101744	0.9193
	HDL (mg/dL)	b5 = 0.063761	r = 0.098495	t = 0.779342	0.4387
	LDL (mg/dL)	b6 = 0.172631	r = 0.388744	t = 3.322289	**0.0015**
	Creatinine (mg/dL)	b7 = 0.267526	r = 0.205048	t = 1.649598	0.1041
	Bilirubin (mg/dL)	b8 = 0.022068	r = 0.028541	t = 0.224826	0.8229
Blood Leukocyte TL (T/S)	Hemoglobin (g/dL)	b1 = 0.003138	r = 0.141452	t = 1.125107	0.2649
	Glycaemia (mg/dL)	b2 = −0.00175	r = −0.089968	t = −0.711294	0.4796
	Cholesterol (mg/dL)	b3 = −0.000016	r = −0.001611	t = −0.012687	0.9899
	Triglycerides (mg/dL)	b4 = 0.001514	r = 0.177891	t = 1.42342	0.1596
	HDL (mg/dL)	b5 = −0.000176	r = −0.008083	t = −0.06365	0.9495
	LDL (mg/dL)	b6 = −0.003492	r = −0.245463	t = −1.993777	**0.0506**
	Creatinine (mg/dL)	b7 = −0.0017	r = −0.039478	t = −0.311094	0.7568
	Bilirubin (mg/dL)	b8 = −0.000485	r = −0.018608	t = −0.146549	0.884

Abbreviations: HDL = high-density lipoprotein, LDL = low-density lipoprotein.

**Table 5 ijms-25-08056-t005:** Multiple regression analysis of the influence of mean HR, HRV parameters (i.e., SDNN, RMSSD), and drugs affecting HRV (i.e., antidepressants, beta-blockers, calcium channel blockers, inhaled or oral beta-mimetics, theophylline, and alpha-adrenergic agonists) on blood leukocyte DNAmAge/TL.

Blood Leukocyte DNAmAge (years)		**b**	**r**	**t Value**	** *p* **
	SDNN	b1 = −0.311272	r = −0.186422	t = −1.564708	0.1223
	RMSSD	b2 = 0.14605	r = 0.114644	t = 0.95165	0.3446
	Mean HR	b3 = −0.403812	r = −0.356467	t = −3.14618	**0.0025**
	Drugs affecting HRV (0 = no; 1 = yes)	b4 = 8.905208	r = 0.297306	t = 2.567761	**0.0124**
Blood Leukocyte TL (T/S)	SDNN	b1 = 0.008358	r = 0.166005	t = 1.388176	0.1696
	MSSD	b2 = −0.006522	r = −0.167858	t = −1.404117	0.1648
	Mean HR	b3 = 0.005025	r = 0.154982	t = 1.293644	0.2002
	Drugs affecting HRV (0 = no; 1 = yes)	b4 = 0.154968	r = 0.176234	t = 1.476368	0.1445

Abbreviations: SDNN = standard deviation of normal-to-normal RR intervals, RMSSD = root mean square of successive RR interval differences, HR = heart rate.

**Table 6 ijms-25-08056-t006:** AgeAcc and predicted TL post-COVID-19 subjects and COPD patients.

Post-COVID-19 Subjects					
N = 17	Age	Blood AgeAcc (years)	IS AgeAcc (years)	Predicted Blood TL (T/S)	Predicted IS TL (T/S)
Mean ± SD	46.00 ± 12.88	−2.59 ± 3.47 *^§^	1.12 ± 4.37 ^§^	1.20 ± 0.06 *^§^	0.93 ± 0.02 ^§^
**COPD patients**					
**N = 7**	**Age**	**Blood AgeAcc (years)**	**IS cells AgeAcc (years)**	**Predicted Blood TL (T/S)**	**Predicted IS TL (T/S)**
Mean ± SD	72.43 ± 6.00	−10.29 ± 3.50 *	−4.29 ± 5.15	1.31 ± 0.03 *	0.97 ± 0.01

Abbreviations: AgeAcc = age acceleration; IS = induced sputum; TL = telomere length. * Paired *t*-test: Post-COVID-19 subjects. Blood AgeAcc versus IS AgeAcc (two-sided *p* = 0.0011). Predicted blood TL (T/S) vs. predicted IS TL (T/S) (two-sided *p* < 0.0001). COPD patients. Blood AgeAcc versus IS AgeAcc (two-sided *p* = 0.0006). Blood TL versus IS TL (two-sided *p* < 0.0001). ^§^ Mann–Whitney U test: Post-COVID-19 blood AgeAcc versus COPD blood AgeAcc (two-sided *p* = 0.0002). Post-COVID-19 IS AgeAcc versus COPD IS AgeAcc (two-sided *p* = 0.012). Post-COVID-19 predicted blood TL (T/S) vs. COPD predicted vlood TL (T/S) (two-sided *p* < 0.0001). Post-COVID-19 predicted IS TL (T/S) vs. COPD predicted IS TL (T/S) (two-sided *p* < 0.0001).

## Data Availability

The datasets presented in this study are available from the corresponding author upon reasonable request.

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
