# Peer review of "Revealing the Hidden Impacts: Insights into Biological Aging and Long-Term Effects in Pauci- and Asymptomatic COVID-19 Healthcare Workers"

_ijms, 2024, doi:10.3390/ijms25158056_

Round 1
Reviewer 1 Report
Comments and Suggestions for Authors
The authors reported the insights into biological aging and long-term effects in COVID-19 healthcare workers. The early molecular markers such as DNA methylation age, telomer length were investigated in blood leucocytes, nasal cells, induced sputum. The manuscript was well written with sufficient data and statistical analysis. However, there are few minor points the authors should adjust before publication.
1) The legend for figures is too small to be able to read. For example, figure 1 with 2 graphs with x and y axis legend too small font size. Please adjust these settings.
2) The manuscript started with table 3, and table 1 and 2 were scatter within different part of the manuscript. This made the reader confused and hard to follow. Please correct the sequence of tables.
3) There were so many table showing details demographics, statistics value which diluted the main text. It’s suggested that selective tables will be kept in main text and moving the rest to the supplemental information.
4) The conclusions were poorly written. Please polish with more findings that were discussed in the main text.
Author Response
REVIEWER REPORT
Reviewer Comments:
Review 1
The authors reported the insights into biological aging and long-term effects in COVID-19 healthcare workers. The early molecular markers such as DNA methylation age, telomer length were investigated in blood leucocytes, nasal cells, induced sputum. The manuscript was well written with sufficient data and statistical analysis. However, there are few minor points the authors should adjust before publication.
We appreciate your positive feedback on our manuscript titled "Revealing the Hidden Impacts: Insights into Biological Aging and Long-Term Effects in pauci- and asymptomatic COVID-19 healthcare workers." We are grateful for your detailed review and suggestions, which have undoubtedly enhanced the quality of our work. Below, we address the specific points you raised.
Comment 1)
The legend for figures is too small to be able to read. For example, figure 1 with 2 graphs with x and y axis legend too small font size. Please adjust these settings.
Answer 1)
We acknowledge the issue with the font size in the figure legends. We have adjusted the font size for Figure 1 and all other figures to ensure that the legends are readable.
Comment 2)
The manuscript started with table 3, and table 1 and 2 were scatter within different part of the manuscript. This made the reader confused and hard to follow. Please correct the sequence of tables.
Answer 2)
Thanks for your comment. We have reorganized the manuscript so that tables are presented in sequential order. Table 1 and Table 2 are now placed before Table 3, following the order of their first mention in the text.
Comment 3)
There were so many table showing details demographics, statistics value which diluted the main text. It’s suggested that selective tables will be kept in main text and moving the rest to the supplemental information.
Answer 3)
Following your suggestion, we recognise the concern regarding the number of tables in the main text. To streamline the manuscript and keep the focus on the main findings, we have moved four detailed demographic and statistical tables to the Supplementary Materials section of the revised manuscript. The number of tables in the main text has been reduced from ten to six. In addition, the table that is now Supplementary Table 1 has been replaced in the main text by the new Figure 1.
Comment 4)
The conclusions were poorly written. Please polish with more findings that were discussed in the main text.
Answer 4)
We have revised the conclusion section to more clearly reflect the key findings discussed in the manuscript. The revised section now highlights the significant correlations and implications of our findings on biological aging and long-term effects in COVID-19 healthcare workers. This specification was added at pages 12-13 lines 589 - 632 of the revised manuscript.
We hope that these revisions adequately address your concerns and improve the clarity and impact of our manuscript. Thank you once again for your valuable feedback.

Reviewer 2 Report
Comments and Suggestions for Authors
The current study titled “Revealing the Hidden Impacts: Insights into Biological Aging and Long-Term Effects in pauci- and asymptomatic COVID-19 healthcare workers” Ref: ijms-3112084, deals with an important subject. Investigation of biological impacts of asymptomatic COVID-19 patients may assist in understanding how the infection can effect health style and biochemical alternations associated with this disease. Although the study is well organized, minor revisions are needed.
- Limited number of patients were considered in this this study.
- No control(s) was considered for this study. At least records of health file(s) of the patients should be considered and discussed revealing previous health conditions/history.
- The conclusion section should be intensively revised revealing the most observed health impacts giving rise to the final conclusion(s) attained of biological aging.
- List of abbreviations is needed.
- Re-arrangement of tables mentioned is needed. How tables 3 and 4 (page 3) were mentioned before table 1 (page 6).
Author Response
REVIEWER REPORT
Reviewer Comments:
Review 2
The current study titled “Revealing the Hidden Impacts: Insights into Biological Aging and Long-Term Effects in pauci- and asymptomatic COVID-19 healthcare workers” Ref: ijms-3112084, deals with an important subject. Investigation of biological impacts of asymptomatic COVID-19 patients may assist in understanding how the infection can effect health style and biochemical alternations associated with this disease. Although the study is well organized, minor revisions are needed.
We appreciate your insightful comments on our manuscript titled "Revealing the Hidden Impacts: Insights into Biological Aging and Long-Term Effects in pauci- and asymptomatic COVID-19 healthcare workers" (Ref: ijms-3112084). We have addressed the specific points you raised as follows:
Comment 1)
Limited number of patients were considered in this this study.
Answer 1)
We acknowledge that the sample size of our study is limited. However, this limitation arises from the low number of HCWs (n=144 out of n=8240 HCWs) infected during the first wave of COVID-19, which is a point of pride for our hospital. Additionally, the number was further reduced by the availability of HCWs who met the inclusion criteria and were willing to participate in the study (n=76). This information can be found on page 9 lines 434-437 and has been also added to the paragraph “3.6 Limitations and Strengths” on pages 8-9 lines 401-405 of the revised manuscript.
Comment 2)
No control(s) was considered for this study. At least records of health file(s) of the patients should be considered and discussed revealing previous health conditions/history.
Answer 2)
We understand the importance of including control data and have indeed reported the lack of a control group as a limitation of our study, as noted on page 8 lines 394-401 of the revised manuscript. Recruiting age-matched controls without COVID-19 was challenging due to the ongoing pandemic, new infections, the start of the vaccination campaign, pandemic-related restrictions, and high workloads. However, we included a comparative analysis with COPD patients as a positive control group for biological aging parameters, particularly lung aging.
Additionally, as suggested by the reviewer, we considered the previous health status and history of the HCWs who participated in the study. This was based not only on the available medical records but also on a rigorously structured questionnaire that assessed the presence of relevant diseases and comorbidities, as well as lifestyle, clinical, biochemical parameters, and occupational history. The information collected by the questionnaire is presented in section "4.2 Information Collected by Questionnaire" on page 10 lines 465 - 483 of the revised manuscript. The results characterizing the study population are presented in Supplementary Table 5.
Comment 3)
The conclusion section should be intensively revised revealing the most observed health impacts giving rise to the final conclusion(s) attained of biological aging.
Answer 3)
We agree that the conclusion section can be improved. We have revised it to more clearly articulate the most observed health impacts and how they lead to the conclusions on biological aging. The revised section now better highlights the significant correlations and implications of our findings. This specification was added at pages 12- 13 lines 589 - 632 of the revised manuscript.
Comment 4)
List of abbreviations is needed.
Answer 4)
A comprehensive list of abbreviations has been added to the manuscript to aid reader comprehension. The list of abbreviations can now be found on page 23 lines 815 - 850 of the revised manuscript.
Comment 5)
Re-arrangement of tables mentioned is needed. How tables 3 and 4 (page 3) were mentioned before table 1 (page 6).
Answer 5)
Thanks for your comment. We have reorganised the tables to follow a logical order. Tables are now presented in the order in which they appear in the text, for a smoother flow and better readability.
We hope these revisions meet your expectations and improve the clarity and impact of our manuscript. Thank you once again for your valuable feedback.

Round 2
Reviewer 1 Report
Comments and Suggestions for Authors
The authors had revised the manuscript and it's ready for publication.